# A Higher Precision Algorithm for Computing the 1-Wasserstein Distance

**Pankaj K. Agarwal**[1*], **Sharath Raghvendra**[2], **Pouyan Shirzadian**[2], **and Rachita Sowle**[2]

[1]Duke University, [2]Virginia Tech

## Abstract

We consider the problem of computing the 1-Wasserstein distance $\mathcal{W}(\mu, \nu)$ between two $d$-dimensional discrete distributions $\mu$ and $\nu$ whose support lie within the unit hypercube. There are several algorithms that estimate $\mathcal{W}(\mu, \nu)$ within an additive error of $\varepsilon$. However, when $\mathcal{W}(\mu, \nu)$ is small, the additive error $\varepsilon$ dominates, leading to noisy results. Consider any additive approximation algorithm with execution time $T(n, \varepsilon)$. We propose an algorithm that runs in $O(T(n, \varepsilon/d) \log n)$ time and boosts the accuracy of estimating $\mathcal{W}(\mu, \nu)$ from $\varepsilon$ to an expected additive error of $\min\{\varepsilon, (d \log_{\sqrt{d}/\varepsilon} n)\mathcal{W}(\mu, \nu)\}$. For the special case where every point in the support of $\mu$ and $\nu$ has a mass of $1/n$ (also called the Euclidean Bipartite Matching problem), we describe an algorithm to boost the accuracy of any additive approximation algorithm from $\varepsilon$ to an expected additive error of $\min\{\varepsilon, (d \log \log n)\mathcal{W}(\mu, \nu)\}$ in $O(T(n, \varepsilon/d) \log \log n)$ time.

## 1 Introduction

Given two discrete probability distributions $\mu$ and $\nu$ whose support $A$ and $B$, respectively, lie inside the $d$-dimensional unit hypercube $[0, 1]^d$ with $\max\{|A|, |B|\} = n$, the 1-*Wasserstein distance* $\mathcal{W}(\mu, \nu)$ (also called the Earth Mover's distance) between them is the minimum cost required to transport mass from $\nu$ to $\mu$ under the Euclidean metric. The special case where $|A| = |B| = n$ and the mass at each point of $A \cup B$ is $1/n$ is called the *Euclidean Bipartite Matching* (EBM) problem. In machine learning applications, one can improve a model $\mu$ by using its Earth Mover's distance from a distribution $\nu$ built on real data. Consequently, it has been extensively used in generative models (Deshpande et al. (2018); Genevay et al. (2018); Salimans et al. (2018)), robust learning (Esfahani & Kuhn (2018)), supervised learning (Luise et al. (2018); Janati et al. (2019)), and parameter estimation (Liu et al. (2018); Bernton et al. (2019)).

Computing the 1-Wasserstein distance between $\mu$ and $\nu$ can be modeled as a linear program and solved in $O(n^3 \log n)$ time (Edmonds & Karp (1972); Orlin (1988)), which is computationally expensive. There has been substantial effort on designing $\varepsilon$-*additive-approximation algorithms* that estimate $\mathcal{W}(\mu, \nu)$ within an additive error of $\varepsilon$ in $n^2 \text{poly}(d, \log n, 1/\varepsilon)$ time (Cuturi (2013); Lin et al. (2019); Lahn et al. (2019)). When $\mathcal{W}(\mu, \nu)$ is significantly smaller than $\varepsilon$, however, the cost produced by such algorithms will be unreliable as it is dominated by the error parameter $\varepsilon$.

To get a higher accuracy in this case, for $\alpha > 0$, one can compute an $\alpha$-*relative* approximation of the 1-Wasserstein distance, which is a cost $w$ that satisfies $\mathcal{W}(\mu, \nu) \leq w \leq \alpha \mathcal{W}(\mu, \nu)$. There has been considerable effort on designing relative-approximation algorithms; however, many such methods suffer from curse of dimensionality; i.e, their execution time grows exponentially in $d$. Furthermore, they rely on fairly involved data structures that have good asymptotic execution times but are slow in practice and difficult to implement, making them impractical (Agarwal & Sharathkumar (2014); Fox & Lu (2020); Agarwal et al. (2022a;b)). The only exception to this is a classical greedy algorithm, based on a $d$-dimensional quadtree, that returns an $O(d \log n)$-relative approximation of the 1-Wasserstein distance in $O(nd)$ time. It has been used in various machine-learning and computer-vision applications (Gupta et al. (2010); Backurs et al. (2020)). In the case of the Euclidean Bipartite

---

*Following convention from Theoretical Computer Science, all authors are ordered alphabetically.

Matching, Agarwal & Varadarajan (2004) and Indyk (2007) generalized the algorithm in the hierarchical framework to achieve a relative approximation ratio of $O(d^2 \log(1/\varepsilon))$ in $\tilde{O}(n^{1+\varepsilon})$ time[1].

In this paper, we design an algorithm that combines any additive approximation algorithm with the hierarchical quad-tree based framework. As a result, our algorithm achieves better guarantees for both additive and relative approximations. To our knowledge, this is the *first* result that combines the power of additive and relative approximation techniques, leading to improvement in both settings.

## 1.1 PROBLEM DEFINITION

We are given two discrete distributions $\mu$ and $\nu$. Let $A$ and $B$ be the points in the support of $\mu$ and $\nu$, respectively. For the distribution $\mu$ (resp. $\nu$), suppose each point $a \in A$ (resp. $b \in B$) has a probability of $\mu_a$ (resp. $\nu_b$) associated with it, where $\sum_{a \in A} \mu_a = \sum_{b \in B} \nu_b = 1$. Let $G(A, B)$ denote the complete bipartite graph where, for any pair of points $a \in A$ and $b \in B$, there is an edge from $a$ to $b$ of cost $\|a - b\|$, i.e, the Euclidean distance between $a$ and $b$.

For each point $a \in A$ (resp. $b \in B$), we assign a weight $\eta(a) = -\mu_a$ (resp. $\eta(b) = \nu_b$). We refer to any point $v \in A \cup B$ with a negative (resp. positive) weight as a *demand point* (resp. *supply point*) with a demand (resp. supply) of $|\eta(v)|$. Given any subset of points $V \subseteq A \cup B$, the weight $\eta(V)$ is simply the sum of the weights of its points; i.e, $\eta(V) = \sum_{v \in V} \eta(v)$. For any edge $(a, b) \in A \times B$, let the cost of transporting a supply of $\beta$ from $b$ to $a$ be $\beta\|a - b\|$. In this problem, our goal is to transport all supplies from supply points to demand points at the minimum cost. More formally, a *transport plan* is a function $\sigma : A \times B \to \mathbb{R}_{\geq 0}$ that assigns a non-negative value to each edge of $G(A, B)$ indicating the quantity of supplies transported along the edge. The transport plan $\sigma$ is such that the total supplies transported into (resp. from) any demand (resp. supply) point $a \in A$ (resp. $b \in B$) is equal to $-\eta(a)$ (resp. $\eta(b)$). The cost of the transport plan $\sigma$, denoted by $w(\sigma)$, is given by $\sum_{(a,b) \in A \times B} \sigma(a, b)\|a - b\|$. The goal of this problem is to find a minimum-cost transport plan.

If two points $a \in A$ and $b \in B$ are co-located (i.e, they share the same coordinates), then due to the metric property of the Euclidean distances, if $\eta(b) = -\eta(a)$, we can match the supplies to the demands at zero cost and remove the points from the input. Otherwise, if $\eta(b) \neq -\eta(a)$, we replace the two points with a single point of weight $\eta(a) + \eta(b)$. By definition, if the weight of the newly created point is negative (resp. positive), we consider it a demand (resp. supply) point. In our presentation, we always consider $A$ and $B$ to be the point sets obtained after replacing all the co-located points. Observe that, after removing the co-located points, the total supply $U = \eta(B)$ may be less than 1. However, it is easy to see that $\eta(B) = -\eta(A)$; i.e, the problem instance defined on $A \cup B$ is *balanced*. We say that a transport plan $\sigma$ is an *$\varepsilon$-close transport plan* if $w(\sigma) \leq \mathcal{W}(\mu, \nu) + \varepsilon U$.

In many applications, the distributions $\mu$ and $\nu$ are continuous or large (possibly unknown) discrete distributions. In such cases, it might be computationally expensive or even impossible to compute $\mathcal{W}(\mu, \nu)$. Instead, one can draw two sets $A$ and $B$ of $n$ samples each from $\mu$ and $\nu$, respectively. Each point $a \in A$ (resp. $b \in B$) is assigned a weight of $\eta(a) = -1/n$ (resp. $\eta(b) = 1/n$). One can approximate the 1-Wasserstein distance between the distributions $\mu$ and $\nu$ by simply solving the 1-Wasserstein problem defined on $G(A, B)$. This special case where every point has the same demand and supply is called the *Euclidean Bipartite Matching* (EBM) problem. A *matching* $M$ is a set of vertex-disjoint edges in $G(A, B)$ and has a cost $1/n \sum_{(a,b) \in M} \|a - b\|$. For the EBM problem, the optimal transport plan is simply a minimum-cost matching of cardinality $n$.

For any point set $P$ in the Euclidean space, let $C_{\max}(P) := \max_{(a,b) \in P \times P} \|a - b\|$ denote the distance of its farthest pair and $C_{\min}(P) := \min_{(a,b) \in P \times P, a \neq b} \|a - b\|$ denote the distance of its closest pair. The *spread* of the point set, denoted by $\Delta(P)$, is the ratio $\Delta(P) = C_{\max}(P)/C_{\min}(P)$. When $P$ is obvious from the context, we simply use $C_{\min}$, $C_{\max}$, and $\Delta$ to denote the distance of its closest and farthest pair and its spread.

## 1.2 RELATED WORK

**Relative Approximations:** In fixed dimensional settings, i.e., $d = O(1)$, there is extensive work on the design of near-linear time Monte-Carlo $(1 + \varepsilon)$-relative approximation al-

---

[1] $\tilde{O}()$ hides $\text{poly}(d, \log n, 1/\varepsilon)$ factors in the execution time.

gorithms for 1-Wasserstein and EBM problems. The execution time of these algorithms are $\Omega(n(d\varepsilon^{-1}\log n)^d)$ (Khesin et al. (2019); Raghvendra & Agarwal (2020); Fox & Lu (2020); Agarwal et al. (2022a)). A recent algorithm presented by Agarwal et al. (2022b) improved the dependence on $d$ slightly and achieved an execution time of $\Omega(n(d\varepsilon^{-1}\log\log n)^d)$. Nonetheless, the exponential dependence on $d$ makes it unsuitable for higher dimensions.

For higher dimensions, a quad-tree based greedy algorithm provides an expected $O(d\log n)$ approximation in $O(nd)$ time. This algorithm constructs a randomly-shifted recursive partition of space by splitting each cell into $2^d$ cells with half the side-length. For every cell of the quad-tree, the algorithm moves excess supply present inside its child to meet any excess demand present inside another child. Agarwal & Varadarajan (2004) combined a different hierarchical partition with an exact solver to get an expected $O(d^2\log 1/\varepsilon)$-approximation in $\tilde{O}(n^{1+\varepsilon})$ time. Indyk (2007) combined the hierarchical greedy framework with importance sampling to estimate the cost of the Euclidean bipartite matching within a constant factor of the optimal. Additionally, there are $\tilde{O}(nd)$ time algorithms that approximate the matching cost with a factor of $O(\log^2 n)$ (Andoni et al. (2008); Chen et al. (2022)).

There are also approximation algorithms that run in $\tilde{O}(n^2)$ time; however, these algorithms rely on several black-box reductions and at present, there are no usable implementations of these algorithms (Agarwal & Sharathkumar (2014); Sherman (2017)). The lack of fast exact and relative approximations that are also implementable have motivated machine-learning researchers to design additive-approximation algorithms, which we discuss next.

**Additive Approximations:** Cuturi (2013) introduced a regularized version of the optimal transport problem that produces an $\varepsilon$-close transport plan, which can be solved using the Sinkhorn method. For input points within the unit hypercube, such an algorithm produces an $\varepsilon$-close transport plan in $\tilde{O}(n^2d/\varepsilon^2)$ time[2](Lin et al. (2019)). One can also adapt graph theoretic approaches including the algorithm by Gabow & Tarjan (1989) to obtain an $\varepsilon$-close solution in $O(n^2\sqrt{d}/\varepsilon + nd/\varepsilon^2)$ time for points within the unit hypercube (Lahn et al. (2019)). Some of the additive approximation methods, including the Sinkhorn method, are highly parallelizable. For instance, the algorithm by Jambulapati et al. (2019) has a parallel depth of $\tilde{O}(1/\varepsilon)$; see also Altschuler et al. (2017; 2019); Blanchet et al. (2018); Quanrud (2018); Dvurechensky et al. (2018); Guo et al. (2020).

### 1.3 Our Results

Let $T(n,\varepsilon)$ be the time taken by an $\varepsilon$-additive-approximation algorithm on an input of $n$ points in the unit hypercube. In Theorem 1.1 and 1.2, we present new algorithms that improve the accuracy of any additive-approximation algorithm for the 1-Wasserstein problem and the EBM problem, respectively.

**Theorem 1.1.** *Given two discrete distributions $\mu$ and $\nu$ whose support lie in the $d$-dimensional unit hypercube and have spread $n^{O(1)}$, and a parameter $\varepsilon > 0$, a transport plan with an expected additive error of $\min\{\varepsilon, (d\log_{\sqrt{d}/\varepsilon} n)\mathcal{W}(\mu,\nu)\}$ can be computed in $O(T(n,\varepsilon/d)\log_{\sqrt{d}/\varepsilon} n)$ time; here, $\mathcal{W}(\mu,\nu)$ is the 1-Wasserstein distance between $\mu$ and $\nu$.*

**Theorem 1.2.** *Given two sets of $n$ points $A$ and $B$ in the $d$-dimensional unit hypercube and a parameter $\varepsilon > 0$, a matching whose expected cost is within an additive error of $\min\{\varepsilon, (d\log\log n)w^*\}$ of the optimal matching cost $w^*$ can be computed in $O(T(n,\varepsilon/d)\log\log n)$ time with high probability.*

Typical additive-approximation algorithms run in $T(n,\varepsilon) = \tilde{O}(n^2)$ time and compute an $\varepsilon$-close transport plan for any arbitrary cost function; i.e, they make no assumption about the distance between points. The inputs to our algorithm, on the other hand, are point sets in the Euclidean space. Therefore, one can use an approximate dynamic nearest neighbor data structure to improve the execution time of such additive-approximation algorithms. In particular, the algorithm by Lahn et al. (2019) runs in $O(1/\varepsilon)$ phases, where each phase executes one iteration of Gabow-Tarjan's algorithm. As shown by Agarwal & Sharathkumar (2014), one can use a $\frac{1}{\sqrt{\varepsilon}}$-approximate dynamic nearest neighbor data structure with a query/update time of $O(n^\varepsilon + d\log n)$ (Andoni et al. (2014))

---

[2]The execution time of Sinkhorn algorithm as well as other additive approximations depend on the diameter $C$ of the point set. In the case of $d$-dimensional unit hypercube, $C = \sqrt{d}$.

to execute each iteration of Gabow-Tarjan's algorithm in $\tilde{O}(n^{1+\varepsilon})$ time. Combining with the algorithms from Theorem 1.1 and 1.2, we obtain the following relative-approximation algorithms. The details are provided in Appendix D[3].

**Theorem 1.3.** *Let $\mu$ and $\nu$ be two discrete distributions whose support lie in the $d$-dimensional unit hypercube and have polynomial spread, and $\varepsilon > 0$ be a parameter. An $O(d/\varepsilon^{3/2})$-approximate transport plan under Euclidean metric can be computed in $O(d^2 n^{1+\varepsilon}/\varepsilon)$ time.*

**Theorem 1.4.** *Given two sets of $n$ points $A$ and $B$ in the $d$-dimensional unit hypercube and a parameter $\varepsilon > 0$, an $O(\frac{d}{\sqrt{\varepsilon}} \log \frac{d}{\varepsilon})$-approximate matching can be computed in $O(dn^{1+\varepsilon} \log \frac{d}{\varepsilon})$ time with high probability.*

In contrast to our results, Agarwal & Varadarajan (2004) compute an $O(d^2 \log \frac{1}{\varepsilon})$-approximate matching in the same time. Therefore, our algorithm computes a more accurate matching for $d > \frac{1}{\sqrt{\varepsilon}}$. For instance, consider the case where $d = \sqrt{\log n}$. For any arbitrarily small constant $\varepsilon > 0$, the algorithm of Theorem 1.4 will run in $\tilde{O}(n^{1+\varepsilon})$ time and return an $O(\sqrt{\log n})$-approximation. In contrast, all previous methods that achieve sub-logarithmic approximation require $\Omega(n^{5/4})$ time (Agarwal & Sharathkumar (2014)).

We also note that all of our algorithms extend to computing transport plans under any $\ell_p$-metric in a straight forward way[4]. For simplicity, we restrict our presentation only to the Euclidean metric.

**Overview of the algorithm:** Our algorithm uses the hierarchical greedy paradigm. In our presentation, we refer to a hypercube as a *cell*. For any cell $\square$, let $V_\square = (A \cup B) \cap \square$. Unlike a quadtree based greedy algorithm, which splits each cell $\square$ into $2^d$ cells, we split it into $\min\{|V_\square|, (4\sqrt{d}/\varepsilon)^d\}$ cells. Thus, the height of the resulting tree $T$ reduces from $O(\log n)$ to $O(\log_{\sqrt{d}/\varepsilon} n)$. For any cell $\square$ of $T$ and any child $\square'$ of $\square$, we move any excess supply or demand inside $\square'$ to its center. Let $\mathcal{A}_\square$ (resp. $\mathcal{B}_\square$) be a set consisting of the center points of all children of $\square$ with excess demand (resp. supply). For any child $\square'$ of $\square$ with excess demand (resp. supply), we assign a weight of $\eta(V_{\square'})$ to its center point in $\mathcal{A}_\square$ (resp. $\mathcal{B}_\square$). Using an additive-approximation algorithm, we compute an $(\varepsilon/d)$-close transport cost between $\mathcal{A}_\square$ and $\mathcal{B}_\square$ in $T(|V_\square|, \varepsilon/d)$ time. We report the sum of the transport costs computed at all cells of $T$ as an approximate 1-Wasserstein distance. This simple algorithm guarantees improvement in the quality of the solutions produced by both additive and relative-approximation algorithms.

From the perspective of relative-approximation algorithms, Agarwal & Varadarajan (2004) as well as Indyk (2007) have utilized a similar hierarchical framework to design approximation algorithms. However, unlike our algorithm, they used an exact solver at each cell that takes $\Omega\left(|\mathcal{A}_\square \cup \mathcal{B}_\square|^3\right)$ time. As a result, to obtain near-quadratic execution time, they divided every cell into $O\left(|V_\square|^{2/3}\right)$ children, i.e., $1/\varepsilon^d \le n^{2/3}$ or $d = O(\log_{1/\varepsilon} n)$. This also forces the height of the tree to be $O(d \log_{1/\varepsilon} n)$, leading to an $O(d^2 \log_{1/\varepsilon} n)$-factor approximation. We replace the $\Omega(n^3)$ time exact solver in their algorithm with a $T(|V_\square|, \varepsilon) = \tilde{O}(|V_\square|^2)$ time additive-approximation algorithm. Therefore, each instance (regardless of the number of non-empty children) can be solved in $\tilde{O}(|V_\square|^2)$ time. As a result, we are able to improve the approximation factor from $O(d^2 \log_{1/\varepsilon} n)$ to $O(d \log_{\sqrt{d}/\varepsilon} n)$ and also remove restrictions on the dimension. Our algorithm now works for any dimension!

**Technical Challenge:** Using an additive-approximation algorithm at each cell increases the error that may be difficult to bound. We use the following observation to overcome this challenge. In Section 2.2, we show that for any point set with spread $\Delta$, an additive-approximation algorithm can be used to compute a 2-relative approximation in $T(n, 1/\Delta)$ time. The algorithm guarantees that the spread of the point set at each cell $\square$ is $O(d/\varepsilon)$, and as a result, we get a 2-relative approximation by using an additive-approximation algorithm in $O(T(n, \varepsilon/d))$ time.

**Improvements for EBM:** For the EBM problem, we obtain an improvement in the approximation ratio, as stated in Theorem 1.2, as follows: Instead of dividing each cell into a fixed number

---

[3]The appendix is provided in the supplemental material.

[4]Owing to a higher complexity of LSH in arbitrary $\ell_p$-metrics, the approximation factor in Theorem 1.3 and 1.4 is slightly higher for arbitrary $\ell_p$-metrics.

$\min\{n, (4\sqrt{d}/\varepsilon)^d\}$ of children, we divide it into $\min\{n, n^{d/2^i}\}$ children at each level $i$; here, level of any cell in the tree is equal to the length of the path from the root to this cell. By doing so, we reduce the height of the tree to $O(\log\log n)$. To analyze the running time, we show that the number of remaining unmatched points over all level $i$ cells is $\tilde{O}(n^{1-\frac{1}{2^i}})$. Since there are only sub-linearly many points remaining, we can afford a larger spread of $O(\frac{d}{\varepsilon}n^{\frac{1}{2^i}})$ and the resulting execution time of the additive approximation will continue to be $O(T(n, \frac{\varepsilon}{d}))$ per level.

**Open Question:** Although our algorithms work in any dimension, we would like to note that the relative approximation factor grows linearly in the dimension. Recently, Chen et al. (2022) removed the dependence on the dimension $d$ from the approximation factor of quad-tree greedy algorithm by using a data dependent weight assignment. It is an open question if their approach can be adapted in our framework leading to a similar improvement in the approximation factor of our algorithms.

## 2 PRELIMINARIES

We begin by introducing a few notations. For a cell $\square$, we denote its side-length by $\ell_\square$ and its *center* by $c_\square$. For a parameter $\ell$, let $\mathbb{G}(\square, \ell)$ denote a grid that partitions $\square$ into smaller cells with side-length $\ell$. Recall that $V_\square$ denotes the subset of $A \cup B$ that lies inside $\square$. We say that $\square$ is *non-empty* if $V_\square$ is non-empty. Recall that $\eta(V_\square) = \sum_{v \in V_\square} \eta(v)$. We define the weight of $\square$ to be $\eta(V_\square)$ and denote it by $\eta(\square)$. We call a cell $\square$ a *deficit cell* if $\eta(\square) < 0$, a *surplus cell* if $\eta(\square) > 0$, and a *neutral cell* if $\eta(\square) = 0$.

In this section, we provide a simple transformation of the input for achieving an additive approximation of the 1-Wasserstein distance. Furthermore, we show how an additive-approximation algorithm for the 1-Wasserstein problem can be used to obtain a $(1+\varepsilon)$-relative-approximation algorithm that runs in $T(n, \varepsilon/\Delta)$ time, where $\Delta$ is the spread of input points.

### 2.1 ADDITIVE APPROXIMATION IN EUCLIDEAN SPACE

For any $\varepsilon > 0$, given an additive-approximation algorithm that runs in $T(n, \varepsilon)$ time, we present an algorithm to compute an $\varepsilon$-close transport cost for distributions inside the $d$-dimensional unit hypercube $\square^*$ in $O(\min\{T(n, \frac{\varepsilon}{2}), n + T((\frac{2\sqrt{d}}{\varepsilon})^d, \frac{\varepsilon}{2})\})$ time.

The algorithm works in two steps. In the first step, it constructs a grid $\mathbb{G} := \mathbb{G}(\square^*, \frac{\varepsilon}{2\sqrt{d}})$ on the unit hypercube $\square^*$ and computes a transport plan $\sigma_1$, as follows. For each non-empty neutral cell of $\mathbb{G}$, $\sigma_1$ arbitrarily transports all supplies to demands within the cell. Similarly, for any deficit (resp. surplus) cell, $\sigma_1$ arbitrarily transports supplies from (resp. to) all supply (resp. demand) points inside the cell to (resp. from) some arbitrary demand (resp. supply) points within the cell.

In the second step, the algorithm constructs a set of demand points $\mathcal{A}$ and a set of supply points $\mathcal{B}$ as follows: For any deficit cell $\square$ (resp. surplus cell $\square'$), the point $c_\square$ (resp. $c_{\square'}$) is added to $\mathcal{A}$ (resp. $\mathcal{B}$) with a weight of $\eta(\square)$ (resp. $\eta(\square')$). Note that $\mathcal{A} \cup \mathcal{B}$ is a balanced instance for the 1-Wasserstein problem. The algorithm computes an $\frac{\varepsilon}{2}$-close transport plan $\sigma_2$ on the instance $\mathcal{A} \cup \mathcal{B}$ in $T(|\mathcal{A}| + |\mathcal{B}|, \varepsilon/2)$ time (See Figure 1). The algorithm returns $w(\sigma_1) + w(\sigma_2)$ as an $\varepsilon$-close transport cost on $A \cup B$.

We provide a discussion on the accuracy of the algorithm in Appendix A. The following lemma follows from the fact that $|\mathcal{A}| + |\mathcal{B}|$ is bounded by $\min\{2n, (2\sqrt{d}/\varepsilon)^d\}$.

**Lemma 2.1.** *Given two point sets $A$ and $B$ in the $d$-dimensional unit hypercube and a parameter $\varepsilon > 0$, an $\varepsilon$-close transport cost can be computed in $O(\min\{T(n, \frac{\varepsilon}{2}), n + T((\frac{2\sqrt{d}}{\varepsilon})^d, \frac{\varepsilon}{2})\})$ time.*

Instead of transporting supplies inside cells arbitrarily, our algorithm in Section 3 recursively applies the same algorithm in each cell and obtains a higher accuracy.

### 2.2 RELATIVE APPROXIMATION FOR LOW SPREAD POINT SETS

In this section, we show that an $\varepsilon$-additive-approximation algorithm can be used to obtain a $(1+\varepsilon)$-relative-approximation algorithm for the 1-Wasserstein problem that runs in $T(n, \varepsilon/\Delta)$ time; here,

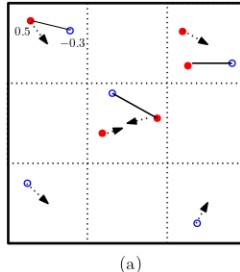 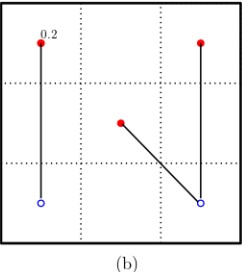

(a)    (b)

Figure 1: (a) The algorithm transports supplies (red disks) to demands (blue circles) within each cell and creates an instance by moving any excess supplies or demands to the center of the corresponding cells, (b) An $\varepsilon/2$-close transport plan is computed on the new problem instance.

$\Delta$ is the spread of the points in $A \cup B$. Since relative approximations are scale invariant, without loss of generality, we assume that the input has $C_{\max} = 1$ and $C_{\min} = 1/\Delta$.

Suppose $-\eta(A) = \eta(B) = U$. The minimum distance between any two points $(a, b) \in A \times B$ is $1/\Delta$. Therefore, the cost of transporting a supply of $U$ is at least $U/\Delta$. We obtain a $(1+\varepsilon)$-relative approximate transport plan by simply executing an additive-approximation algorithm with an additive error of $U(\varepsilon/\Delta)$ in time $T(n, \varepsilon/\Delta)$.

**Lemma 2.2.** *Given two point sets $A$ and $B$ in the d-dimensional unit hypercube with a spread of $\Delta$, a $(1+\varepsilon)$-approximate transport plan can be computed in $T(n, \varepsilon/\Delta)$ time.*

# 3   AN $O(d \log_{\sqrt{d}/\varepsilon} n)$-APPROXIMATION ALGORITHM FOR 1-WASSERSTEIN PROBLEM

In this section, we present our algorithm that satisfies the bounds presented in Theorem 1.1. We begin by defining a hierarchical partitioning and a tree $T$ associated with the hierarchical partitioning. Each node in $T$ corresponds to a non-empty cell in our hierarchical partition, and we do not distinguish between the two. We partition each cell of side-length $\ell$ into $\lceil 4\sqrt{d}/\varepsilon \rceil^d$ cells of side-length at most $(\varepsilon/4\sqrt{d})\ell$. We construct our hierarchical partition in a randomly-shifted fashion as follows.

**Hierarchical Partitioning:** First, we pick a point $\xi$ uniformly at random from the unit hypercube $[0, 1]^d$ and set $\square^* = [-1, 1]^d + \xi$. Note that $\square^*$ is a hypercube of side-length 2 containing all points in $A \cup B$. We designate $\square^*$ as the root of $T$. Let $\kappa = \lceil 4\sqrt{d}/\varepsilon \rceil$. For any cell $\square$, if only one point of $A \cup B$ lies inside $\square$, we designate $\square$ as a leaf cell in $T$. For any non-leaf cell $\square$, we construct its children by partitioning $\square$, using a grid $\mathbb{G}_\square = \mathbb{G}(\square, \ell_\square/\kappa)$, into $\lceil 4\sqrt{d}/\varepsilon \rceil^d$ cells and create a child node for each non-empty cell of this grid. We denote the set of children of a non-leaf cell $\square$ by $\mathsf{C}[\square]$. Assuming the spread of $A \cup B$ is $n^{O(1)}$, the height of $T$, denoted by $h$, is $O(\log_{\sqrt{d}/\varepsilon} n)$.

Similar to quadtrees, our hierarchical partitioning can be seen as a sequence of grids $\langle \mathbb{G}_0, \mathbb{G}_1, \ldots, \mathbb{G}_h \rangle$, where $\mathbb{G}_0$ is the root and each grid $\mathbb{G}_i$ refines the cells of grid $\mathbb{G}_{i-1}$. For each grid $\mathbb{G}_i$, we denote the cell-side-length of $\mathbb{G}_i$ by $\ell_i$. Next, we describe our algorithm.

**Computing an approximate 1-Wasserstein distance:** In this section, we present our algorithm for computing an approximate cost of an optimal transport plan on $A \cup B$.

*1 - Creating an instance at each cell:* For each cell $\square$ of $T$, we create a balanced instance $\mathcal{I}_\square$ of the 1-Wasserstein problem as follows. If $\square$ is a leaf cell, then it contains a single point $u$ of $A \cup B$, which we add to $\mathcal{I}_\square$. In addition, we add the center point $c_\square$ with a weight $-\eta(\square)$ to $\mathcal{I}_\square$. Otherwise, $\square$ is a non-leaf cell. For any child $\square' \in \mathsf{C}[\square]$, if $\square'$ is a deficit or surplus cell, we add $c_{\square'}$ with a weight $\eta(\square')$ to $\mathcal{I}_\square$. The weight $\eta(\square')$ represents the excess demands or supplies of the points in $V_{\square'}$. Furthermore, we add the center point $c_\square$ with a weight $-\eta(\square)$ to $\mathcal{I}_\square$. The instance $\mathcal{I}_\square$ is balanced and has a spread of $O(d/\varepsilon)$.

*2 - Estimating the 1-Wasserstein distance:* In this step, for the root cell $\square^*$, we compute an $\varepsilon/d$-close transport plan $\sigma_{\square^*}$ on $\mathcal{I}_{\square^*}$. Furthermore, for each cell $\square$ at any level $i > 0$, using the

algorithm from Lemma 2.2, we compute a 2-approximate transport plan $\sigma_\square$ on $\mathcal{I}_\square$. Our algorithm then reports the total cost of the transport plans computed at all cells of $T$, i.e, $w := \sum_{\square \in T} w(\sigma_\square)$, as an approximate 1-Wasserstein distance.

**Retrieving an approximate transport plan:** We retrieve an approximate transport plan $\sigma$ on the point set $A \cup B$ by processing grids $\langle \mathbb{G}_h, \dots, \mathbb{G}_0 \rangle$ in decreasing order of their level. First, for each non-empty cell $\square$ of $\mathbb{G}_h$, $\square$ is a leaf cell and $V_\square$ contains only one point. We map the only point in $V_\square$ to $c_\square$. For some $i < h$, assume (inductively) that after processing the non-empty cells of the grid $\mathbb{G}_{i+1}$, the following conditions (i)–(iii) hold for the current transport plan $\sigma$ within any cell $\square$ in $\mathbb{G}_{i+1}$: (i) if $\square$ is a neutral cell, then $\sigma$ transports every supply to some demand point inside $\square$, (ii) if $\square$ is a deficit (resp. surplus) cell, then $\sigma$ transports all supplies (resp. demands) inside $\square$ to (resp. from) some demand (resp. supply) point within $\square$, and, (iii) if $\square$ is a deficit (resp. surplus) cell, the excess demand (resp. supply) is mapped to $c_\square$. Given this, we show how to process any non-empty cell $\square$ of $\mathbb{G}_i$ so that (i)–(iii) holds for $\square$.

Recollect that $\sigma_\square$ is a transport plan computed by our algorithm on $\mathcal{I}_\square$. By condition (iii), the excess supplies or demands at any child $\square'$ of $\square$ is mapped to $c_{\square'}$. Therefore, for any pair of children $\square_1, \square_2 \in \mathsf{C}[\square]$, where $\square_1$ is a surplus cell and $\square_2$ is a deficit cell, the transport plan $\sigma$ transports $\sigma_\square(c_{\square_1}, c_{\square_2})$ supplies from $c_{\square_1}$ to $c_{\square_2}$. In addition, for any child $\square_1$ (resp. $\square_2$) of $\square$, suppose $\square_1$ (resp. $\square_2$) is a surplus (resp. deficit) cell. If $\sigma_\square(c_{\square_1}, c_\square) > 0$ (resp. $\sigma_\square(c_{\square_2}, c_\square) > 0$), then we map the supplies (resp. demands) from $c_{\square_1}$ (resp. $c_{\square_2}$) to $c_\square$. It is easy to confirm that after processing $\square$, (i)–(iii) holds for $\square$. From triangle inequality, $w(\sigma)$ is upper-bounded by the total cost of the transport plans computed for each cell of $T$; i.e, $w(\sigma) \leq \sum_{\square \in T} w(\sigma_\square)$.

**Efficiency:** For any $i$, let $\mathsf{C}_i$ denote the set of non-empty cells of $T$ at level $i$. For each cell $\square \in \mathsf{C}_i$, let $n_\square$ be the number of points in $\mathcal{I}_\square$. Since the spread of the points in $\mathcal{I}_\square$ is $O(d/\varepsilon)$, executing the algorithm from Lemma 2.2 on $\mathcal{I}_\square$ takes $O(T(n_\square, \varepsilon/d))$ time (the same bound holds for the root cell as well). Since $\mathcal{I}_\square$ contains at most one point for each non-empty child of $\square$, $n_\square \leq \min\{|V_\square|, (4\sqrt{d}/\varepsilon)^d\}$. Therefore, $\sum_{\square \in \mathsf{C}_i} n_\square \leq \sum_{\square \in \mathsf{C}_i} |V_\square| = n$. Since $T(n, \varepsilon) = \Omega(n)$, the running time of our algorithm on cells at level $i$ is $O(\sum_{\square \in \mathsf{C}_i} T(n_\square, \varepsilon/d)) = O(T(n, \varepsilon/d))$. Summing over all levels, the running time of our algorithm is $O(T(n, \varepsilon/d) \log_{\sqrt{d}/\varepsilon} n)$.

When the dimension is a small constant, we get an improved running time as follows. For each level $i$ of $T$, there are at most $n$ non-empty cells at level $i$ and the instance created at each cell has a size of at most $(4\sqrt{d}/\varepsilon)^d$. Therefore, $O(\sum_{\square \in \mathsf{C}_i} T(n_\square, \varepsilon/d)) = O(nT((4\sqrt{d}/\varepsilon)^d, \varepsilon/d)) = n(\frac{d}{\varepsilon})^{O(d)}$ and the overall running time will be improved to $n(\frac{d}{\varepsilon})^{O(d)} \log_{\sqrt{d}/\varepsilon} n$.

**Quality of Approximation:** In this part, we analyze the approximate 1-Wasserstein distance computed by our algorithm. First, we show that the reported cost is $\varepsilon$-close. For the root cell $\square^*$, our algorithm computes an $\varepsilon/d$-close transport plan $\sigma_{\square^*}$. The remaining demands and supplies are recursively transported within the children of $\square^*$, where each child has diameter $\varepsilon/2$. Therefore, similar to Section 2.1, we can argue that our algorithm reports an $\varepsilon$-close 1-Wasserstein distance. Next, we show that the reported cost is an $O(d \log_{\sqrt{d}/\varepsilon} n)$-approximation of the 1-Wasserstein distance.

For each level $i < h$, we show that the expected cost of the transport plans computed for all cells of level $i$ is $\mathbb{E}\left[\sum_{\square \in \mathsf{C}_i} w(\sigma_\square)\right] = O(d)w(\sigma^*)$. Here, $\sigma^*$ is an optimal transport plan on $A \cup B$, $\mathsf{C}_i$ denotes the set of non-empty cells of $T$ at level $i$, and the expectation is over the choice of the random shift of the hierarchical partitioning. We bound $\mathbb{E}\left[\sum_{\square \in \mathsf{C}_i} w(\sigma_\square)\right]$ in two steps as follows. In the first step, we assign a budget to every edge $(a, b)$ with $\sigma^*(a, b) > 0$ and show that the total budget assigned to all such edges is (in expectation) $O(d)w(\sigma^*)$. In the second step, we redistribute this budget to the cells of level $i$ in a way that the budget received by any cell $\square$ is at least $w(\sigma_\square)/2$. Summing over all $O(\log_{\sqrt{d}/\varepsilon} n)$ levels of $T$, the expected value of the total cost computed at all levels $i < h$ is $O(d \log_{\sqrt{d}/\varepsilon} n)w(\sigma^*)$.

Additionally, we show that the expected cost of mapping all points to the centers of the cells of level $h$ is $O(d \log_{\sqrt{d}/\varepsilon} n)w(\sigma^*)$. Details of each step is provided in Appendix B. Theorem 1.1 follows from combining the two bounds.

## 4    AN $O(d \log \log n)$-APPROXIMATION ALGORITHM FOR EBM PROBLEM

Suppose $A$ and $B$ are two sets of $n$ points inside the $d$-dimensional unit hypercube, where each point $a \in A$ (resp. $b \in B$) has a weight $\eta(a) = -1/n$ (resp. $\eta(b) = 1/n$). In this section, we present an approximation algorithm for the EBM problem satisfying the bounds claimed in Theorem 1.2. Note that by invoking Lemma 2.1 on $A \cup B$, one can compute an $\varepsilon$-close transport cost on $A \cup B$. To boost the accuracy of the algorithm of Lemma 2.1, in this section, we present an $O(d \log \log n)$-approximation algorithm for the EBM problem. To satisfy the bounds claimed in Theorem 1.2, one can then report the minimum of the costs computed by the two algorithms.

**Input transformation:** We transform input points such that (1) all coordinates are positive integers bounded by $n^{O(1)}$, (2) an optimal matching on the transformed points is a $(1 + \varepsilon)$-approximate matching with respect to the original points, and (3) the cost of the optimal matching is $O(d^{3/2} n \log n / \varepsilon)$. Similar transformations have been applied in several papers in the literature (Agarwal et al. (2017); Lahn & Raghvendra (2021)). We describe this transformation in Appendix C.1. As before, we can match and remove any co-located points $a \in A$ and $b \in B$.

Next, assuming $T(n, \varepsilon) = O(\frac{n^k}{\varepsilon} \text{poly}(d, \log n, \log \frac{1}{\varepsilon}))$ for some $k \geq 1$, we describe our EBM algorithm. Our algorithm is easily adaptable to use any additive-approximation algorithm with a running time of $T(n, \varepsilon) = O(n^k \varepsilon^{-t} \text{poly}(d, \log n, \log \frac{1}{\varepsilon}))$, where $k \geq 1$ and $t$ is a fixed constant.

**Overview of the algorithm:** Similar to Section 3, our EBM algorithm constructs a hierarchical partitioning and the associated tree $T$, and executes the algorithm from Lemma 2.2 on the instance created for each cell of $T$. In contrast to Section 3, the tree $T$ constructed by our EBM algorithm has height $O(\log \log n)$, resulting in an improved approximation factor. The hierarchical partitioning of this section differs from the one in Section 3 in two ways. First, we partition the root cell into a grid $\mathbb{G}_1$ with cell-side-length of $\Theta(d^{5/2} n \log n / \varepsilon^2)$ at the first level. The grid $\mathbb{G}_1$ may result in a high branching factor at the root; however, we show that, with probability at least $1 - \varepsilon / \sqrt{d}$, no edges of an optimal matching will cross $\mathbb{G}_1$. Therefore, with that probability, all cells of $\mathbb{G}_1$ are neutral cells and the problem instance for the root is an empty instance; i.e, the branching factor of the root will not impact the running time of our algorithm. Second, for any cell $\square$ of level $i$, instead of splitting $\square$ into a fixed number $\min\{n, (4\sqrt{d}/\varepsilon)^d\}$ of children, we divide $\square$ into $\min\{n, n^{d/2^i}\}$ cells. Although this results in a spread of $\tilde{O}(n^{1/2^i})$ for the problem instance $\mathcal{I}_\square$, we show that the expected number of remaining unmatched points over all cells of level $i$ is $\tilde{O}(n^{1-1/2^i})$. Therefore, the total execution time of our algorithm remains $T(n, \varepsilon/d)$ per level. These modifications result in a tree $T$ of height $O(\log \log n)$. We describe the details below.

**Hierarchical Partitioning:** Define $\delta := 5d^2 \varepsilon^{-1} \log n$. Similar to Section 3, we define a cell $\square^*$ as a randomly-shifted hypercube that contains all points of $A \cup B$ and has a side-length of $2 \max\{C_{\max}, \ell_1/\varepsilon\}$, where $\ell_1 = \frac{\sqrt{d}}{\varepsilon} \delta n$. We designate $\square^*$ as the root of $T$ ($\square^*$ is at level 0 of $T$). Define a grid $\mathbb{G}_1 := \mathbb{G}(\square^*, \ell_1)$. We add each non-empty cell of $\mathbb{G}_1$ to the tree as the children of $\square^*$. We construct the hierarchical partitioning in a recursive fashion as follows. For any non-root cell $\square$ of $T$, if $\square$ contains only one point of $A \cup B$, then we designate $\square$ as a leaf cell. Otherwise, let $\square$ be a cell of level $i$. Define the grid $\mathbb{G}_\square = \mathbb{G}(\square, \delta n^{1/2^i})$ and add the non-empty cells of $\mathbb{G}_\square$ to $T$ as the children of $\square$. For simplicity in presentation, we assume $n^{1/2^i}$ is an integer. For any cell $\square$, denote the set of children of $\square$ in $T$ by $\mathsf{C}[\square]$. The height of $T$, denoted by $h$, is $O(\log \log n)$.

Similar to Section 3, our hierarchical partitioning is also a sequence of grids $\langle \mathbb{G}_0, \mathbb{G}_1, \ldots, \mathbb{G}_h \rangle$, where $\mathbb{G}_0$ is the root cell $\square^*$, $\mathbb{G}_1$ has a cell-side-length of $\ell_1 = \frac{\sqrt{d}}{\varepsilon} \delta n$, and for each $2 \leq i \leq h$, the cell-side-length of $\mathbb{G}_i$ is $\ell_i = \delta n^{1/2^{i-1}}$.

**Computing an approximate matching cost:** To estimate the matching cost, similar to Section 3, our algorithm creates an instance of the 1-Wasserstein problem for each cell of the tree $T$. Using the algorithm from Lemma 2.2, our algorithm computes a 2-approximate transport plan for the instance created for each cell and returns the total cost of such transport plans as an approximate matching cost. This completes the description of the algorithm.

We describe the details of retrieving a matching in Appendix C.2, the quality of approximation in Appendix C.3, and the efficiency of our algorithm in Appendix C.4.

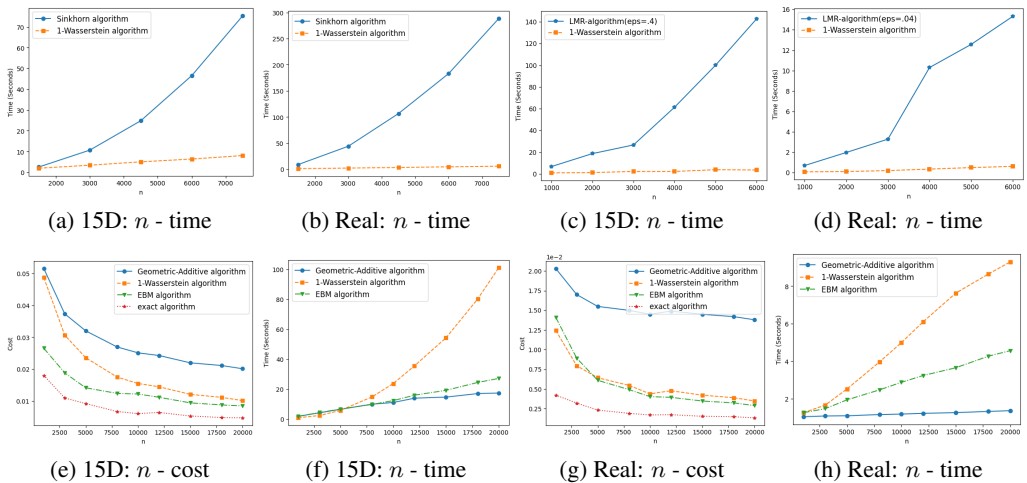

|   |   |   |   |
|---|---|---|---|
| (a) 15D: $n$ - time | (b) Real: $n$ - time | (c) 15D: $n$ - time | (d) Real: $n$ - time |
| (e) 15D: $n$ - cost | (f) 15D: $n$ - time | (g) Real: $n$ - cost | (h) Real: $n$ - time |

Figure 2: (a) and (b) Comparison with the Sinkhorn algorithm, (c) and (d) Comparison with the LMR algorithm, and (e)–(h) Comparison with the Geometric-Additive algorithm

## 5 EXPERIMENTS

In this section, we conduct experiments to show that our algorithms from Section 3 and Section 4 improve the accuracy of the additive approximation algorithms. We test an implementation of our algorithm, written in Python, on discrete probability distributions derived from real-world and synthetic data sets. All tests are executed on a computer with a 2.50 GHz Intel Core i7 processor and 8GB of RAM using a single computation thread.

**Datasets:** We test our algorithms on two sets of $n$ samples taken from synthetic (distribution) and real-world data sets. For each dataset, we use our algorithms to compute the minimum-cost matching between these samples and present our results averaged over 10 executions. To generate the synthetic data, we sample from a uniform distribution inside a 2-dimensional unit square placed on a random plane in 15-dimensional space (15D). For a real-world dataset, we use the *Adult Census Data* (UCI repository), which is a point cloud in $\mathbb{R}^6$ with continuous features for $35,000$ individuals, divided into two categories by income (Dua & Graff (2017)). See Appendix E for the results of our experiments on additional datasets.

**Results:** In our first experiment, we compare our algorithm with existing additive approximation schemes, namely the Sinkhorn method (Cuturi (2013)) (resp. LMR algorithm (Lahn et al. (2019))). We use the Sinkhorn (resp. LMR) algorithm as a black-box within our algorithm from Section 3 (1-Wasserstein algorithm) and compare its execution time with the standard Sinkhorn (resp. LMR) implementation. We set the parameters of the Sinkhorn and LMR algorithms in such a way that the error produced by them matches with the error produced by our 1-Wasserstein algorithm. As shown in Figure 2 (a)–(d), our algorithm runs significantly faster than both the Sinkhorn and the LMR algorithms while producing solutions of similar quality. In our second experiment, we also compare the accuracy as well as the computation time of our algorithms (1-Wasserstein algorithm and our EBM algorithm from Section 4) with the additive approximation algorithm from Section 2.1 (Geometric-Additive algorithm). For this experiment, our algorithms use the Sinkhorn algorithm as a black-box. The results of this experiment are shown in Figure 2 (e)–(h). We observe that our algorithms achieves a better accuracy (see Figure 2 (a) and (c)), especially on real-world data sets. As expected, the execution time of our algorithms increase slightly (see Figure 2 (d)). Furthermore, we observe that as the sample size increases, the costs returned by our algorithms converge to the optimal cost, whereas the Geometric-Additive Approximation based approach does not.

Finally, we highlight the scalability of our algorithms. For an input of 3 million points drawn from a 2 dimensional uniform distribution on the unit square, our 1-Wasserstein algorithm runs in 593 seconds and computes an approximate transport cost of 0.0093. Furthermore, for an input of 1.5 million points for the 15D dataset, our 1-Wasserstein algorithm computes an approximate cost of 0.0304 in 608 seconds.

## ACKNOWLEDGEMENT

We would like to acknowledge, Advanced Research Computing (ARC) at Virginia Tech, which provided us with the computational resources used to run the experiments. Research presented in this paper was funded by NSF grants CCF-1909171, CCF-2223871, IIS-1814493, CCF-2007556, and CCF-2223870. We would like to thank the anonymous reviewers for their useful feedback.

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
