# OpenReview forum: "A Higher Precision Algorithm for Computing the $1$-Wasserstein Distance"
_ICLR.cc/2023/Conference — ICLR 2023 notable top 25%_

### Official Review · Reviewer_4W22 · 2022-10-20

**Confidence:** 2
**Correctness:** 4
**Technical Novelty And Significance:** 3
**Empirical Novelty And Significance:** Not applicable
**Recommendation:** 8

**Clarity, Quality, Novelty And Reproducibility:**

The paper is well-written on its own. However, it does not present the existing techniques very clearly. As the paper states that it combines the existing techniques from the additive error algorithms and the relative error algorithms, it is not very clear to people who have not worked directly on the problem what the new innovation is and what the difficulty is to combine the techniques. It is thus difficult to assess the merits of the paper for someone who has not worked directly on the problem.

[Given the short review period, I have not been able to check the proofs carefully or read the proofs in the appendix.]

Minor points:
- Line 5 of abstract: add a comma after “dominates”
- Line 8: additive factor -> additive error (it’s additive and thus cannot be a factor)
- Section 1, paragraph 2, line 4: poly{...} -> poly(...). The same typo appears multiple times throughout the paper.
- Section 1, paragraph 2, line 5: there is no limiting process here, why “converge”?
- Section 1.1, paragraph 1, last line: change the semicolon to a comma
- Page 4, line -18: it is better to write $d = O(log_{1/\epsilon} n)$
- Page 4, line -7: “Since the spread…” This has never appeared before this point, it’s better to say that the algorithm guarantees the spread is …
- Page 4, line -7: the spread is $O(d/\epsilon)$, not exactly $d/\epsilon$, right? The same typo appears multiple times in the paper.
- Page 7, paragraph starting with “In the next section”: In the next section -> below (it’s still in the same section)
- Page 7, paragraph starting with “Quality of Approximation”: please fix “in this section”
- Page 8, paragraph starting with “Hierarchical Partitioning”, line 7: the footnote mark 4 is confusing, as it can be interpreted as the fourth power.
- Page 9, paragraph starting with “Datasets”, line 1: real data -> real-world data
- Page 9, paragraph starting with “Datasets”, line 7: add a comma before “which”
- Page 9, Figure 2: Colours are hard to distinguish when printed black & white. It’s better to use different line markers.


**Strength And Weaknesses:**

Strengths: Improves the (expected) additive error when the Wasserstein distance is smaller than $\epsilon/(d\log_{\sqrt{d}/\epsilon} n)$.

~~Weaknesses: The main weakness to me is that the guarantee is in the expected additive error, not a worst-case error guarantee. (The abstract should also state that it’s the “expected” additive error clearly.)~~

[Update] The authors have clarified that repeating $O(\log(1/\delta))$ times gives a worst-error guarantee of the same magnitude with probability at least $1-\delta$.

**Summary Of The Paper:**

The paper considers approximating the 1-Wasserstein distance between two discrete distributions up to an additive error. Suppose that $\mu$ and $\nu$ are two discrete distributions contained within the unit hypercube. Both are supported on n points and both have a $\operatorname{poly}(n)$ spread. The paper shows a randomised algorithm approximating the 1-Wasserstein distance $W_1(\mu,\nu)$ up to an expected additive error of $\min\\{\epsilon, (d\log_{\sqrt{d}/\epsilon} n)W_1(\mu,\nu) \\}$ in $O(T(n,\epsilon/d)\log_{\sqrt{d}/\epsilon} n)$ time, where $T(n,\epsilon)$ is the time taken by an $\epsilon$-additive approximation algorithm. Based on this core result, the paper also obtains relative-error algorithms for the Euclidean bipartite matching problem.

**Summary Of The Review:**

This is a solid paper on its own. ~~but it is difficult to evaluate the novelty in the techniques as the paper is not clear on this aspect. The guarantee on expected additive error is not enticing, either.~~

[Update] The techniques are mostly based on the existing ones but it is nontrivial to combine them with small twists in various places. I would recommend acceptance though it looks more like an ESA paper instead of an ICLR paper.

---

> ### Author Response · Authors · 2022-11-18
> **Response to the Reviewer 4W22**
>
> We appreciate your thoughtful review and constructive feedback.
> We have revised our submission based on your comments. We also address your main comments below.
>
> **Comment:** *``The main weakness to me is that the guarantee is in the expected additive error, not a worst-case error guarantee."*
>
> **Our Response:** As stated in the common response, for the $1$-Wasserstein (resp. EBM) problem, our algorithm guarantees a *worst-case* additive error of $\varepsilon$. When $\mathcal{W}(\mu,\nu)$ is significantly smaller than $\varepsilon$, our algorithm can improve the additive error from $\varepsilon$ to an $\alpha$-relative approximation where the expected value of $\alpha$ is $O(d\log_{\sqrt{d}/\varepsilon}n)$ (resp. $O(d\log\log n)$).
>
> We can obtain this improvement to $O(d\log_{\sqrt{d}/\varepsilon}n)$ relative approximation with a probability of $1-1/n$ by simply executing our algorithm $O(\log n)$ times and returning the smallest approximation.
>
> **Comment:** *``The paper does not present the existing techniques very clearly."*
>
> **Our Response:** In our submission, we have presented a comparison on our methods with existing techniques in Section 1.3 ``Overview of our algorithm''. To further address your concern, we have also included a brief description of the hierarchical greedy paradigm within the related work section.
>
>
> **Comment:** *``What the new innovation is and what the difficulty is to combine the techniques."*
>
> **Our Response:** We highlight our innovations in our common response and provide some additional details that may be useful to understand our contributions better.
>
> Typical combinatorial exact and relative approximation algorithms for the $1$-Wasserstein problem run $\Omega(\sqrt{n})$ phases, each taking $O(n^2)$ time. We highlight our novel observations that leads us to a better relative approximation algorithm.
> 1) We observe that by using *any* state-of-the-art additive approximation algorithm, we can obtain a relative approximation algorithm that executes only $\Delta$ phases; here $\Delta$ is the spread of the point set (Lemma 2.2).
> 2) We then define a hierarchical decomposition scheme with a branching factor of $\min\\{n, (\sqrt{d}/\varepsilon)^d\\}$. For each cell, we create an instance of $1$-Wasserstein problem on the centers of its non-empty children. Note that the spread $\Delta$ of each instance is only $d/\varepsilon$. Using observation 1, each of these instances can be solved quickly and our algorithm simply returns the sum of these distances computed at each cell (Theorem 1.1 and 1.3). The quality of the solution produced by this algorithm depends linearly on the height of the tree, i.e., $O(\log_{\sqrt{d}/\varepsilon}n)$.
> 3) For the EBM problem,  we obtain an improvement in the relative approximation as follows. We create a new hierarchical structure, where
> instead of dividing each cell into a fixed number of children, we divide each cell of level $1$ into level $2$ cells of side-length $n^{1/2}$, a cell of level $2$ into level $3$ cells of side-length $n^{1/4}$, and so on. As a result, the height of the tree reduces from $O(\log_{\sqrt{d}/\varepsilon} n)$ to $O(\log\log n)$. Similar to the $1$-Wasserstein problem, our algorithm uses this hierarchical structure to decompose the EBM computation into several $1$-Wasserstein computations, each of which is solved using an additive approximation algorithm. Due to the reduced height of this hierarchical decomposition, the quality of the solution improves to $O(d\log \log n)$ (Corollary C.2 in the appendix). Note that the instances created for cells that are closer to the root have a high spread. As a result, the number of phases executed to solve such instances becomes large. However, to bound the running time, we show that these instances have a sub-linear size (Lemma C.4 in the appendix), and therefore, the time taken to execute each phase becomes sub-quadratic. Thus, the increase in the number of phases is balanced by the reduction in the time taken for executing each phase leading to a similar running time as our algorithm for the $1$-Wasserstein problem (Section C.4 in the appendix).

---

> > ### Comment · Reviewer_4W22 · 2022-11-19
> > **Thanks for your response**
> >
> > > We can obtain this improvement ... with a probability of $1-1/n$ by simply executing our algorithm $O(\log n)$ times and returning the smallest approximation.
> >
> > Oh yes, that is a good point. Please state this explicitly in your revision that repeating $O(\log(1/\delta))$ times gives a success probability at least $1-\delta$. That clarifies my main concern and I have raised my score.
> >
> > Thanks for the detailed explanation and updating the manuscript. The technical innovation is now clearer.

---

### Official Review · Reviewer_LSEt · 2022-10-28

**Confidence:** 3
**Correctness:** 4
**Technical Novelty And Significance:** 3
**Empirical Novelty And Significance:** Not applicable
**Recommendation:** 8

**Clarity, Quality, Novelty And Reproducibility:**

The ideas are presented very clearly, and the paper is well-written. Previously known techniques are extended in a novel way. Experiments are performed on synthetic datasets and should be reproducible.

**Strength And Weaknesses:**

Strengths:
- This paper makes an improvement on an important problem and at the same time gives both an additive and relative approximation guarantee, which could make the algorithm even more widely applicable.
- The paper is quite well-written and easy to follow.

Weaknesses:
- Many of the core ideas used are taken from prior work. However, the paper does seem to have technical depth on its own.



**Summary Of The Paper:**

In this paper, the authors consider the problem of computing the 1-Wasserstein distance (Earth Mover’s distance) between two d-dimensional discrete distributions $\nu$ and $\mu$. In this problem the mass distributed over a set of supply nodes according to distribution $\nu$, needs to be transported and distributed to the demand nodes in $\mu$. The amount of mass supplied or demanded by each node is given and the per unit transportation cost between a pair of nodes is proportional to the $\ell_2$ (Euclidean) distance between the nodes. The goal is to compute the minimum cost required to transport the mass from the supply nodes to demand nodes. A special case of this problem where all nodes have the same amount of demanded or supplied mass, is also considered. This problem is called Euclidean Bipartite Matching (EBM), since in the optimal solution a demand node will be connected to a single supply node and vice versa. While prior work on this problem had provided both additive and relative approximation guarantees, the contribution of this paper is an algorithm that combines the two types of guarantees. Specifically, for the Earth Mover’s distance problem, the error is the minimum between $\varepsilon$ and $(d\log_{\sqrt{d}/\varepsilon}n) \mathcal{W}(\mu,\nu)$, where $\mathcal{W}(\mu,\nu)$ is the optimal cost. The second term provides an improved relative approximation compared to $d\log n$ achieved in prior work, while the additive error is bounded by $\varepsilon$ at the same time.

Similarly to prior work, the algorithm uses a randomly shifted hierarchical partitioning of the space recursively subdividing cells with multiple points in them. The algorithm first tries to satisfy the demands using the supply from the same cell of the partitioning and this is also done in a recursive manner, therefore starting from the lower levels of the hierarchical tree and moving up. When a particular cell has more supply than demand (or vice versa), the remainder is moved to its center to be handled by a higher level cell. This paper differentiates from prior work on relative approximation algorithms by using exact solvers for each cell, whereas faster approximate solvers are used in this paper leading to smaller cell size and better approximation guarantees. For the problem of Euclidean Bipartite Matching (EBM), the authors exploit the fact that cells in the hierarchical tree are more likely to be balanced (in terms of internal supply and demand) and they are able to improve the relative approximation to $d\log log n$, while maintaining the additive approximation.


**Summary Of The Review:**

I believe this paper makes a solid contribution by providing improved additive and relative approximation guarantees with a single algorithm for a well-known problem.

---

> ### Author Response · Authors · 2022-11-18
> **Response to the Reviewer LSEt**
>
> We appreciate your thorough review and feedback. Thank you for acknowledging the technical depth in our paper. We have restated the main contributions of our paper as well as how it differentiates from existing work in the common response.

---

### Official Review · Reviewer_8UAQ · 2022-10-30

**Confidence:** 3
**Correctness:** 4
**Technical Novelty And Significance:** 4
**Empirical Novelty And Significance:** 4
**Recommendation:** 8

**Clarity, Quality, Novelty And Reproducibility:**

The algorithms in the paper combine known techniques but the end result is novel enough.

**Strength And Weaknesses:**

+ The paper makes good progress on an important problem.
+ The algorithms are interesting and novel.
- The experimental results are not so impressive.

**Summary Of The Paper:**

This paper gives a new algorithm for computing the Earth Mover's Distance between two discrete distributions in the hypercube. This is an extensively studied problem with several theoretical algorithms papers studying it. This present paper exploits some of their efficient-in-practice techniques in order to design a practical algorithm.

**Summary Of The Review:**

I recommend accepting the paper.

---

> ### Author Response · Authors · 2022-11-18
> **Response to the Reviewer 8UAQ**
>
> Thank you for your thorough review and helpful feedback.
>
> **Comment:**  *``The experimental results are not so impressive.''*
>
> **Our Response:** We thank the reviewer for raising this issue. In the original submission, because of lack of space, many important experimental results were included in the Appendix. We have now moved two of these experiments to the main text and highlight them below.
>
> 1) We compared our algorithm with existing additive approximation schemes, namely the Sinkhorn method (resp. LMR algorithm [1]). We use the Sinkhorn (resp. LMR) algorithm as a black-box within our $1$-Wasserstein algorithm and compare its execution time with the standard Sinkhorn (resp. LMR) implementation. We set the parameters of the Sinkhorn and LMR algorithms in such a way that the error produced by them matches with the error produced by our $1$-Wasserstein algorithm. As shown in Figure 2 (a)-(d), our algorithm runs significantly faster than both the Sinkhorn and the LMR algorithms while producing solutions of similar quality.
>
> 2) We also highlight the scalability of our algorithms. For an input of $3$ million points drawn from a $2$ dimensional uniform distribution on the unit square, our $1$-Wasserstein algorithm runs in $593$ seconds and computes an approximate transport cost of $0.0093$. Furthermore, for an input of $1.5$ million points for the $15D$ dataset, our $1$-Wasserstein algorithm computes an approximate cost of $0.0304$ in $608$ seconds.
>
>
> [1] Nathaniel Lahn, Deepika Mulchandani, and Sharath Raghvendra. A graph theoretic additive
> approximation of optimal transport. Advances in Neural Information Processing Systems,
> 32, 2019.

---

### Author Response · Authors · 2022-11-18
**General Response to All Reviewers**

We thank the reviewers for their insightful comments, which have helped in improving the presentation of the paper. We thank PC for giving us an opportunity to respond to the comments. We begin by addressing a couple of high-level comments made by multiple reviewers and then respond to comments by individual reviewers.

There are two high-level comments made by Reviewers 8UAQ and 4W22.

**Technical contributions**


1) Our algorithm is the *first* result that combines the power of additive and relative approximation algorithms, leading to improvement in both settings. Our algorithm in Sections $3$ (resp. Section $4$) computes the $1$-Wasserstein distance (resp. EBM) $\mathcal{W}(\mu,\nu)$ with a *worst-case* additive error of  at most $\varepsilon$. If $\mathcal{W}(\mu,\nu)$ is significantly smaller than $\varepsilon$, our algorithm can improve the additive error from $\varepsilon$ to an $\alpha$-relative error (i.e, error being $\alpha\mathcal{W}(\mu, \nu)$), where the expected value of $\alpha$ is $O(d\log_{\sqrt{d}/\varepsilon}n)$ (resp. $O(d\log\log n)$).

2)  Our algorithm obtains significantly better performance using the following novel ideas:
    - If input points have a low spread (ratio of the maximum to minimum pairwise distance), we can use *any* state-of-the-art additive approximation algorithm to obtain a significantly faster relative approximation algorithm for the $1$-Wasserstein distance (Section 2.2).
     - We also show how the $1$-Wasserstein distance for any point set can be estimated (within an $O(d\log_{\sqrt{d}/\varepsilon}n)$-factor) by the sum of several $1$-Wasserstein distances between point sets of small spread. Our algorithm from Section $3$ follows by combining this decomposition with our faster algorithm for low spread point sets.
      - For the EBM problem, we describe a novel *adaptive hierarchical partitioning scheme* that computes (within an improved relative $O(d\log\log n)$-approximation) the EBM on the original input as the union of the EBM on several subsets of the input. We prove that each such subset either has a sub-linear size or has a small spread. Therefore, the EBM instance on each of these subsets can be solved quickly leading to a significant improvement in the approximation quality without any loss in the execution time.


**Experimental results:**
Our experimental results show that:
1) Our algorithm runs significantly faster than both the Sinkhorn algorithm and the LMR algorithms [1] while producing solutions of similar quality, and

 2) Our algorithm can process very large point sets. For instance, for an input of $1.5$ million points for the 15D dataset, our $1$-Wasserstein algorithm runs in $608$ seconds.


[1] Nathaniel Lahn, Deepika Mulchandani, and Sharath Raghvendra. A graph theoretic additive
approximation of optimal transport. Advances in Neural Information Processing Systems,
32, 2019.

---

### Comment · Area_Chair_kM1j · 2022-11-18
**Could you show the relatedness to the recent works?**

Dear Authors,

Thank you for your response. Recently, data-dependent approaches have been proposed for approximating the 1-Wasserstein distance.  So, could you show the relationship between your method and the following works? In addition to the reviews, this information is highly useful for discussion.

- New streaming algorithms for high dimensional EMD and MST, STOC 2022 https://dl.acm.org/doi/abs/10.1145/3519935.3519979
- Approximating 1-Wasserstein distance with trees, TMLR 2022 https://openreview.net/forum?id=Ig82l87ZVU
- Budget-Constrained Bounds for Mini-Batch Estimation of Optimal Transport, arXiv 2022 https://arxiv.org/abs/2210.13630
- Learning Ultrametric Trees for Optimal Transport Regression, arXiv 2022 https://arxiv.org/abs/2210.12288

The bottom 2 papers are not published yet;  But, it would be great to discuss if they are related to your paper.

Thanks,

AC

---

> ### Author Response · Authors · 2022-11-27
> **Response to AC**
>
> Thank you for your question about contrasting our paper with [1, 2, 3, 4]. Papers [2, 3, 4] are primarily experimental in nature and do not provide any provable guarantees on the quality of the transport plan they compute. On the other hand, while [1] provides a rigorous theoretical analysis, it does not provide an experimental analysis of its approach. Our algorithm has both a provable relative approximation as well as an experimental analysis.
>
>
> **Comparison with [1]:** The result in [1] uses a data-dependent weight assignment on edges to improve the relative approximation of the classical quad-tree greedy algorithm from $O(d\log n)$ to $O(\log n\log \log n)$. Our algorithm uses a different hierarchical structure with an adaptive branching factor to  improve the approximation factor from $O(d\log n)$ (of quad-tree greedy) to $O(d \log \log n)$. Our algorithm produces a better quality approximation than [1] when $d < \log n$. An intriguing and challenging open question is whether the data-dependent weight assignment of [1] can be combined with our hierarchical structure leading to $O(\log \log n)$ approximation for matchings in any dimension.
>
> **Comparison with [2, 4]:**
> Similar to [1], the papers [2, 4] are based on embedding the distances to a quad-tree $T$ based metric and assigning data-dependent weights to the edges of $T$. These papers primarily differ in how they compute these weights on the edges of $T$.  Note that [2, 4] do not provide any provable bounds on the quality of approximations produced by their algorithm.
>
> Unlike [2, 4] that embed costs into tree metric, we embed the costs into a different hierarchical structure with an adaptive branching factor. Our embedding is not a tree embedding since it allows for edges between every pair of siblings. Our embedding in combination with the observations from our common response lead to a strong relative approximations of OT.
>
>
> **Comparison with [3]:**
> [3] suggests approximating the OT-cost with the so-called *Batch-Hierarchical OT* (BHOT) and presents a relative approximation to BHOT. In BHOT, one divides the input points $A$ and $B$ into $k$ equal-sized mini-batches and creates a complete bipartite graph on them. The weight of an edge from a mini-batch of $A$ to a mini-batch of $B$ is equal to the OT-cost between them. The BHOT-cost is simply the weight of the minimum-cost matching in this bipartite graph. We are not aware of any upper bound on the ratio of BHOT-cost to the original OT-cost. Thus, unlike our paper, this paper does not give any guarantees on the approximation ratio with respect to the original OT-cost.
>
> [3] approximates the BHOT-cost by embedding the mini-batches into $\ell_1$-space and then using a variant of the quad-tree greedy algorithm.  It is a challenging open question to determine if, instead of quad-tree greedy approach, an adaptive partitioning scheme like the one in our paper can lead to a better approximations of BHOT.
>
>
>
>
>
>
> **References**
>
> [1] Xi Chen, Rajesh Jayaram, Amit Levi, and Erik Waingarten. New streaming algorithms for high dimensional emd and mst. In Proceedings of the 54th Annual ACM SIGACT Symposium on Theory of Computing, pages 222–233, 2022.
>
> [2] Makoto Yamada, Yuki Takezawa, Ryoma Sato, Han Bao, Zornitsa Kozareva, and Sujith Ravi. Approximating 1-wasserstein distance with trees. arXiv preprint arXiv:2206.12116, 2022.
>
> [3] David Alvarez-Melis, Nicol`o Fusi, Lester Mackey, and Tal Wagner. Budget-constrained bounds for mini-batch estimation of optimal transport. arXiv preprint arXiv:2210.13630, 2022.
>
> [4] Samantha Chen, Puoya Tabaghi, and Yusu Wang. Learning ultrametric trees for optimal transport regression. arXiv preprint arXiv:2210.12288, 2022.

---

> > ### Comment · Area_Chair_kM1j · 2022-11-29
> > **Thanks**
> >
> > Dear authors,
> >
> >  Thank you for the clarification. The response is very useful for discussion.
> >
> > AC

---

### Decision · Program_Chairs · 2023-01-20

**Decision:**

Accept: notable-top-25%

**Justification For Why Not Higher Score:**

Wasserstein distance approximation is a very important topic in ML and the paper proposes a new Wasserstein distance approximation algorithm. However, this paper has a bit narrower focus than the typical Wasserstein distance approximation and optimal transport problems. Thus, I recommend this for the spotlight paper.

**Justification For Why Not Lower Score:**

This is an important work for 1-Wasserstein distance approximation.

**Metareview: Summary, Strengths And Weaknesses:**

In this paper, the authors propose an approximation algorithm for the 1-Wasserstein distance between two discrete distributions up to an additive error. The proposed method is novel and all the reviewers are happy to accept the paper. So, I also vote for acceptance.

In the final version, please clearly discuss the connection of the proposed method with the recent work including "New streaming algorithms for high dimensional EMD and MST, STOC 2022 https://dl.acm.org/doi/abs/10.1145/3519935.3519979".



**Note From Pc:**

if the above contains the word "oral" or "spotlight" please see: "oral" presentation means -> notable-top-5% and "spotlight" means -> notable-top-25%. As stated in our emails, we are disassociating presentation type from AC recommendations

**Summary Of Ac-Reviewer Meeting:**

This is not a borderline paper.